# Iron availability and oxygen tension regulate the *Yersinia Ysc* type III secretion system to enable disseminated infection

Diana Hooker-Romero[1], Erin Mettert[2], Leah Schwiesow[3], David Balderas[1], Pablo A. Alvarez[1], Anadin Kicin[1], Azuah L. Gonzalez[1], Gregory V. Plano[4], Patricia J. Kiley[2], Victoria Auerbuch[1]*

1 Department of Microbiology and Environmental Toxicology, University of California Santa Cruz, Santa Cruz, CA United States of America, 2 Department of Biomolecular Chemistry, University of Wisconsin-Madison, Madison, WI, United States of America, 3 Department of Molecular, Cell, and Developmental Biology, University of California Santa Cruz, Santa Cruz, CA, United States of America, 4 Department of Microbiology and Immunology, University of Miami, Miami, FL, United States of America

☯ These authors contributed equally to this work.
* vastone@ucsc.edu

**Data Availability Statement:** All relevant data are within the manuscript and its Supporting Information files.

## Abstract

The enteropathogen *Yersinia pseudotuberculosis* and the related plague agent *Y. pestis* require the Ysc type III secretion system (T3SS) to subvert phagocyte defense mechanisms and cause disease. Yet type III secretion (T3S) in *Yersinia* induces growth arrest and innate immune recognition, necessitating tight regulation of the T3SS. Here we show that *Y. pseudotuberculosis* T3SS expression is kept low under anaerobic, iron-rich conditions, such as those found in the intestinal lumen where the *Yersinia* T3SS is not required for growth. In contrast, the *Yersinia* T3SS is expressed under aerobic or anaerobic, iron-poor conditions, such as those encountered by *Yersinia* once they cross the epithelial barrier and encounter phagocytic cells. We further show that the [2Fe-2S] containing transcription factor, IscR, mediates this oxygen and iron regulation of the T3SS by controlling transcription of the T3SS master regulator LcrF. IscR binds directly to the *lcrF* promoter and, importantly, a mutation that prevents this binding leads to decreased disseminated infection of *Y. pseudotuberculosis* but does not perturb intestinal colonization. Similar to *E. coli*, *Y. pseudotuberculosis* uses the Fe-S cluster occupancy of IscR as a readout of oxygen and iron conditions that impact cellular Fe-S cluster homeostasis. We propose that *Y. pseudotuberculosis* has coopted this system to sense entry into deeper tissues and induce T3S where it is required for virulence. The IscR binding site in the *lcrF* promoter is completely conserved between *Y. pseudotuberculosis* and *Y. pestis*. Deletion of *iscR* in *Y. pestis* leads to drastic disruption of T3S, suggesting that IscR control of the T3SS evolved before *Y. pestis* split from *Y. pseudotuberculosis*.

## Author summary

The *Yersinia* type III secretion system (T3SS) is an important virulence factor of the enteropathogen *Yersinia pseudotuberculosis* as well as *Yersinia pestis*, the causative agent of

**Funding:** This study was supported by National Institutes of Health (www.NIH.gov) grant R01AI119082 (to VA and PJK) and the Ford Foundation (www.fordfoundation.org) (to DHR). The funders had no role in study design, data collection and analysis, decision to publish, or preparation of the manuscript.

**Competing interests:** The authors have declared that no competing interests exist.

plague. Although the T3SS promotes *Yersinia* survival in the host, its activity is not compatible with bacterial growth. Therefore, *Yersinia* must control where and when to express the T3SS to optimize fitness within the mammalian host. Here we show that *Yersinia* sense iron availability and oxygen tension, which vary between the intestinal environment and deeper tissues. Importantly, we show that eliminating the ability of *Y. pseudotuberculosis* to control its T3SS in response to iron and oxygen does not affect colonization of the intestine, where the T3SS is dispensable for growth. However, loss of T3SS control by iron and oxygen severely decreases disseminated infection. We propose that *Y. pseudotuberculosis* senses iron availability and oxygen tension to detect crossing the intestinal epithelial barrier. As the mechanism by which iron and oxygen control the T3SS is completely conserved between *Y. pseudotuberculosis* and *Y. pestis*, yet *Y. pestis* is not transmitted through the intestinal route, we propose that *Y. pestis* has retained this T3SS regulatory mechanism to suit its new infection cycle.

## Introduction

Iron availability and oxygen concentration play important roles in bacterial growth and gene regulation. Iron (Fe) ions serve as metabolic cofactors necessary for crucial processes such as respiration, oxidative stress resistance, and virulence factor production [1]. Iron is able to adopt two stable valences, ferric and ferrous, providing iron-containing proteins with a considerable oxidation-reduction potential [2]. The level of iron required for optimal bacterial growth is $\sim 10^{-6}$ M. However, the level of free iron in mammalian tissues is typically $\sim 10^{-18}$ M [3]. This is because in healthy mammals, iron is mostly found inside cells associated with heme or metalloproteins, or stored in ferritin. Furthermore, the trace amounts of extracellular iron are bound to the high affinity glycoproteins lactoferrin and transferrin [2, 3]. During infection, the host immune system withholds Fe from invading pathogens even more stringently in a process called nutritional immunity. For example, expression of lactoferrin, found in bodily secretions and neutrophil granules, is induced during an inflammatory response and acts to sequester iron [4]. In addition, neutrophils in the inflamed intestine produce lipocalin-2, an antimicrobial protein that captures enterobactin, the primary siderophore of many enteric bacteria [5]. The importance of nutritional immunity is exemplified by the increased susceptibility of individuals with iron overload, including β-thalassemia, sideroblastic anemia, and hemochromatosis, to systemic infections with pathogenic *Yersinia*, *Vibrio parahaemolyticus*, and several other pathogens [6, 7].

Most bacterial pathogens have evolved an extensive armory of uptake systems tailored to different iron sources and availability. To ensure optimal iron uptake, pathogenic *Yersinia* have numerous iron uptake and detoxification mechanisms preferentially activated depending on the environment and iron source they encounter. For example, when oxygen is present, yersiniabactin is upregulated to acquire $Fe^{3+}$ iron [8]; when oxygen is absent, the $Fe^{2+}$ iron uptake systems Yfe and Feo are upregulated [9]. However, in most bacteria, iron levels are homeostatically regulated through repression of these iron uptake systems to avoid the deleterious effects of excess iron. This global iron-dependent regulation is largely controlled by the repressor Fur (Ferric Uptake Regulator), which represses target gene expression under iron-replete conditions [10]. Other studies in Enterobacteriaceae have also demonstrated that many iron responsive genes, including *fur* itself, can be regulated by other transcription factors, such as OxyR, in response to environmental changes such as oxidative stress or siderophore uptake [2, 11, 12]. This co-regulation of iron-dependent genes ensures that pathogens use the most

fitting iron uptake system during their passage through the complex environment of the host organism and maintain iron homeostasis.

In addition to the challenge of acquiring sufficient iron for growth in the face of nutritional immunity, enteropathogens must adapt and respond to changes in oxygen availability within mammalian tissues [13]. For example, the healthy intestinal lumen is anaerobic, while the concentration of oxygen increases closer to the intestinal crypts [14]. A number of bacterial pathogens sense fluctuations in oxygen concentration to control virulence factors such as the type III secretion system (T3SS), which serves to inject effector proteins into target host cells to modulate host defenses. In the case of *Shigella flexneri*, it was shown that the absence of oxygen increased the number of T3SS needles formed on the bacterial surface; however, active secretion of the effector protein Ipa did not occur until oxygen was present. This suggests that the *Shigella* T3SS is primed in the anaerobic environment of the intestinal lumen and becomes active when the bacterium approaches the epithelium where oxygen diffuses from capillaries and T3SS activity is required for invasion and intracellular growth [15]. Furthermore, a recent study in *S. enterica* serovar Typhimurium suggested that T3SS expression is tightly regulated by the small RNAs FnrS and ArcZ such that the T3SS is expressed at an intermediate oxygen concentration predicted to be found near the intestinal epithelium [16]. In enterohaemorrhagic *E. coli*, T3SS activity is upregulated under anaerobic conditions when either nitrate or TMAO are used as alternative terminal electron acceptors, although the mechanism behind this remains unclear [17]. While *Yersinia* type III secretion was suggested to be controlled by oxygen [18], the mechanism and underlying physiological relevance is unknown.

One transcription factor known to be controlled by both iron and oxygen is IscR, whose DNA-binding activity is regulated by its association with an iron-sulfur cluster [2Fe-2S][19–22]. IscR functions as a sensor of cellular Fe-S cluster status, which in turn is influenced by iron availability and oxidative stress. IscR regulates Fe-S cluster biogenesis mediated by the housekeeping Isc pathway and the stress-responsive Suf pathway. The [2Fe-2S] cluster holo-form of IscR binds to two distinct DNA motifs (type I and type II). In contrast, the clusterless apo-IscR form binds only type II motifs. In *E. coli*, IscR represses its own expression through binding to two type I motifs in the *isc* promoter, enabling high iron and low oxygen conditions to decrease overall IscR levels [23]. In addition to controlling its own transcription, IscR is also a global regulator of gene expression. For example, in *E. coli* and *Vibrio vulnificus*, IscR controls expression of many genes involved in Fe-S biogenesis and other cellular processes, such as motility, adhesion to host cells, and hemolytic activity [21]. Recently, IscR was shown to be required for type III secretion and virulence in the food-borne pathogen *Yersinia pseudotuberculosis* [24]. In this organism, IscR binds to a type II motif in the promoter controlling expression of the T3SS master regulator, LcrF, suggesting that either holo or apo-IscR can promote LcrF expression [24]. However, how iron and/or oxygen might control the expression of LcrF and the T3SS through IscR is unknown. In addition, as over 200 genes are differentially expressed in the presence or absence of IscR in *Y. pseudotuberculosis* [24], it is likely that the virulence attenuation of the *Y. pseudotuberculosis* Δ*iscR* mutant stems from both lack of T3SS expression as well as misregulation of other genes that contribute to virulence. Therefore, the importance of IscR control of LcrF and the *Yersinia* T3SS remains ill-defined.

A recent study in *Salmonella enterica* serovar Typhimurium suggested that IscR leads to induction of the SPI-1 T3SS in response to iron sufficiency. Considering the SPI-1 T3SS is required for *Salmonella* to invade intestinal epithelial cells, this suggests that *Salmonella* uses IscR, in conjunction with other factors such as those mentioned above, to sense the gut environment, upregulate the SPI-1 T3SS, and cross the gut barrier [22]. In contrast, enteropathogenic *Yersinia* need their T3SS *after* crossing the intestinal barrier. In this current study we determine the environmental conditions that affect IscR regulation of the *Yersinia* T3SS. Our

data suggests that *Y. pseudotuberculosis* uses IscR to sense when it has crossed the intestinal barrier and needs its T3SS to evade host defenses. This is in contrast to what is observed for *Salmonella* IscR control of the SPI-1 T3SS, but in agreement with the course of enteropathogenic *Yersinia* infection.

## Results

### IscR binding to the *lcrF* promoter is critical for type III secretion

To investigate the environmental cues that impact IscR in *Yersinia*, we constructed a mutation in the *lcrF* promoter predicted to disrupt IscR binding (Fig 1A, *lcrF*$^{pNull}$) [25]. As IscR is a global regulator in *Yersinia* [24], the *lcrF*$^{pNull}$ mutant should only be defective in T3SS control, while a Δ*iscR* mutant displays additional T3SS-independent phenotypes (see below). As predicted, purified *E. coli* apo-IscR bound the WT *lcrF* promoter region but not the mutated *lcrF*$^{pNull}$ promoter (Fig 1B), indicating that IscR binding was ablated by this mutation. Consistent with our previous results showing that IscR is essential for type III secretion under standard aerobic conditions [24], the *lcrF*$^{pNull}$ mutant was as deficient as the Δ*iscR* mutant in Yop secretion (Fig 1C). To determine if the *lcrF*$^{pNull}$ mutation affected bacterial replication, we performed growth assays at different temperatures. As expected, at the environmental temperature of 26˚C, the *lcrF*$^{pNull}$ mutant displayed the same growth rate as both the wild type (WT) strain and a strain lacking the pYV plasmid encoding the T3SS (pYV⁻). However, at the mammalian body temperature of 37˚C, the *lcrF*$^{pNull}$ mutant grew at a rate in between that of WT and the pYV⁻ mutant (S1 Fig). This is consistent with our previous results showing that the Δ*iscR* mutation partially rescues the growth restriction associated with active type III secretion [24]. Therefore, although the *lcrF*$^{pNull}$ mutant retains a wildtype IscR allele, it is defective in the IscR-LcrF-T3SS pathway, making it suitable for assessing the role of IscR in direct regulation of the T3SS.

### Oxygen induces Ysc T3SS expression

To examine the effect of oxygen on the regulation of *Yersinia iscR* and the Ysc T3SS, *Y. pseudotuberculosis* was grown under either aerobic or anaerobic conditions, and T3SS expression and activity was analyzed. Importantly, the WT, Δ*iscR*, and *lcrF*$^{pNull}$ strains displayed similar growth under anaerobic conditions at 26˚C, with a doubling time of ~2 hours (Fig 2A). After shifting cultures to 37˚C, we observed that *iscR* mRNA levels were induced 38-fold (p<0.0001) under aerobic conditions relative to anaerobic conditions (Fig 2B), consistent with holo-IscR repression of the *isc* operon anaerobically [26]. Similarly, *lcrF* expression was induced 138-fold (p<0.0001) in response to oxygen. In addition, IscR and LcrF protein levels as well as T3SS activity, as measured by observing relative secreted levels of the T3SS effector protein YopE, were higher under aerobic conditions than anaerobic conditions (Fig 2C), consistent with previously published data [18]. Taken together, these data demonstrate that IscR, LcrF, and T3SS expression is induced in the presence of oxygen and that LcrF expression correlates with IscR expression.

Since IscR has been shown to differentially regulate gene expression in response to iron-limiting conditions in *E. coli* and *Pseudomonas aeruginosa* [19, 27, 28], we sought to determine the effect of iron on the expression of both *iscR* and T3SS genes in *Yersinia*. We previously developed a protocol for limiting *Y. pseudotuberculosis* for iron and demonstrated induction of a Fur-repressed gene [29], indicating efficacy of iron limitation. To mimic host temperature and activate expression of the T3SS, iron-starved *Y. pseudotuberculosis* was incubated at 37˚C under aerobic and anaerobic conditions in media that contained either 6.58 μM FeSO$_4$ (iron-replete) or 0.0658 μM FeSO$_4$ (iron-limited). qPCR analysis revealed that expression of *iscR*

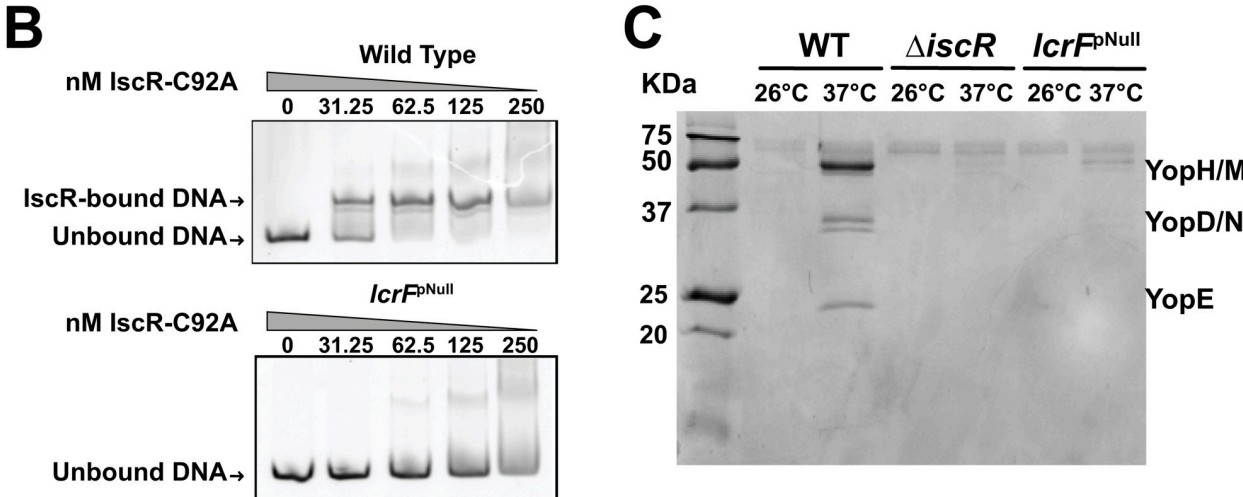

**Fig 1. Mutation of the identified IscR binding site in the *lcrF* promoter disrupts apo-IscR binding and type III secretion. (A)** Nucleotide sequences of the IscR binding site in the WT strain (top) and the *lcrF*^pNull^ mutant (bottom). Bases previously shown to be important for IscR binding are in bold [25]. Mutated residues in the *lcrF*^pNull^ mutant are underlined. **(B)** Electrophoretic mobility shift assays (EMSAs) were performed on DNA fragments (217 bp) containing the WT and *lcrF*^pNull^ mutant promoter sequences upstream of the *yscW-lcrF* operon. Concentrations of *E. coli* apo-IscR, afforded by the IscR-C92A mutant protein, are denoted above the gel lanes. One representative experiment out of three is shown. **(C)** *Y. pseudotuberculosis* was grown in low calcium 2xYT media containing iron and the T3SS was induced at 37˚C for 2 hours. T3SS cargo proteins secreted into culture supernatant and precipitated with TCA were visualized with Coomassie blue. One representative experiment out of three is shown.

mRNA levels did not significantly change in response to iron availability under aerobic conditions, indicating that IscR is largely in the apo-form under aerobic conditions as it is in *E. coli* (Fig 3A). Similarly, IscR protein levels were similar in iron-replete and iron-limited aerobic conditions (Fig 3B) consistent with the lack of negative autoregulation of *iscR* expression by apo-IscR. While *lcrF* expression was decreased by ~1.4-fold under iron-limited conditions (Fig 3A, p = 0.0005), this decrease did not lead to a change in LcrF protein levels (Fig 3C) nor any significant change in expression (Fig 3A and 3D) or secretion (Figs 3C and S2) of the LcrF target genes *yopE* and *yopH*. However, the Δ*iscR* and the *lcrF*^pNull^ mutants displayed a significant decrease in *lcrF* and *yopE* mRNA levels (Fig 3A), *yopH* expression (Fig 3D), and LcrF, YopE, and YopD protein levels compared to WT (Fig 3C), consistent with apo-IscR playing a major role in regulation of the *Yersinia* T3SS under aerobic conditions [30]. Collectively, these data show that under aerobic conditions, *Yersinia* IscR and LcrF expression as well as T3SS activity are not affected by iron availability, consistent with the control of T3SS being mediated by apo-IscR.

## Iron limitation induces type III secretion under anaerobic conditions

In order to test whether iron availability affects IscR, LcrF, and T3SS expression in the absence of oxygen, iron-limited *Yersinia* were incubated anaerobically without any alternative terminal electron acceptors for 12 hrs. Following extended growth under iron-limited conditions, the doubling times of the WT, Δ*iscR*, and *lcrF*^pNull^ strains were similar at 37˚C in iron-replete versus limited media (~10 hours; S3 Fig). In contrast to our observations in the presence of

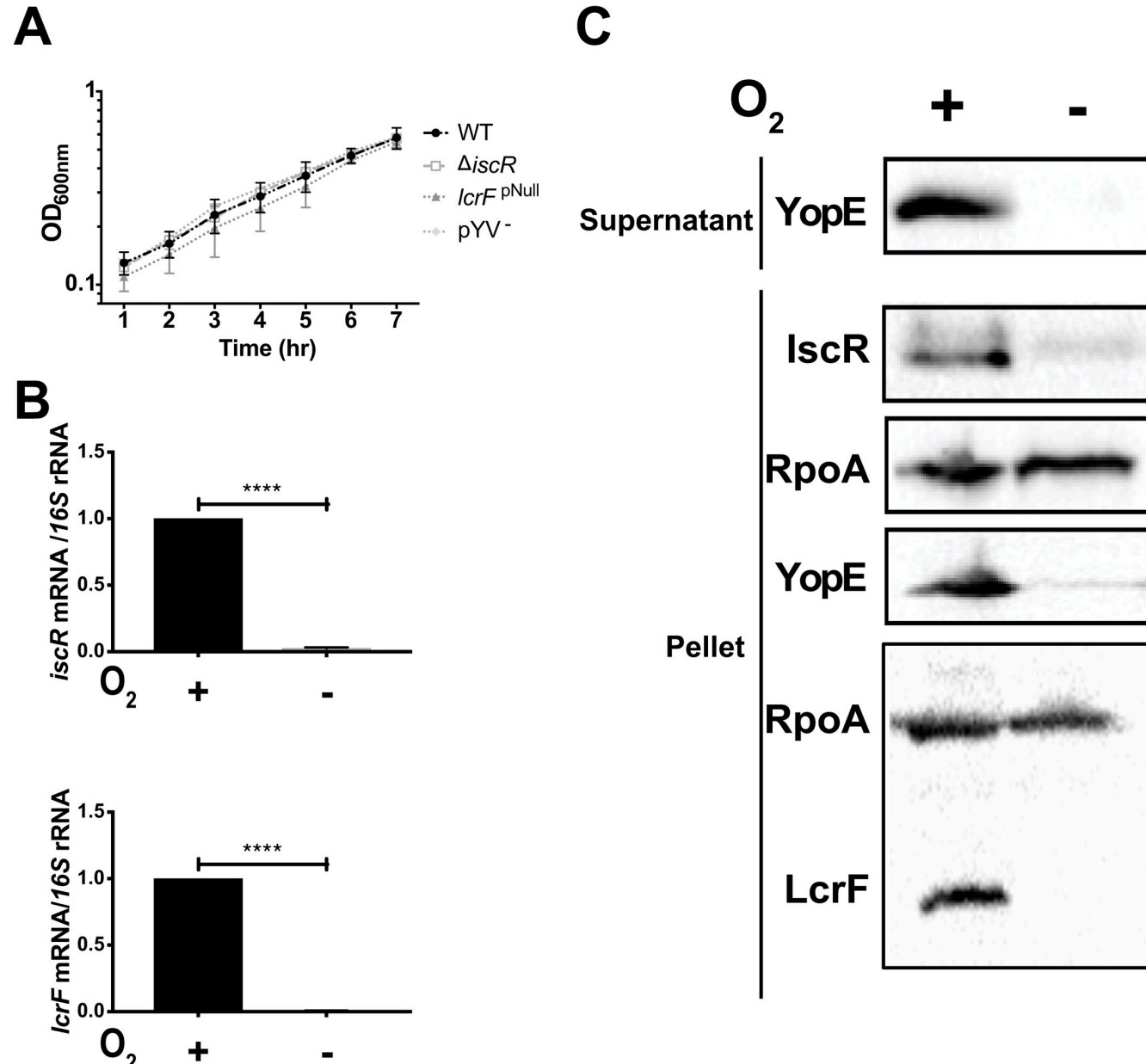

**Fig 2. The *Yersinia* Ysc type III secretion system is induced by oxygen.** (A) *Y. pseudotuberculosis* was grown under anaerobic, iron-replete conditions at 26°C and optical density at 600 nm (OD$_{600nm}$) measured every hour. *Y. pseudotuberculosis* pYV⁻ lacks the pYV virulence plasmid and serves as a T3SS-deficient control strain. Data shown represent an average of three biological replicates. **(B-C)** *Y. pseudotuberculosis* WT was grown in M9/high glucose (iron-replete) in the presence or absence of oxygen and type III secretion induced by shifting to 37°C for 4 hrs. **(B)** RNA was extracted from the bacterial cultures and *iscR* and *lcrF* gene expression analyzed using qPCR. Relative mRNA levels for each gene of interest were normalized to 16S rRNA levels. Data shown represent the average of three independent experiments. **** p ≤ 0.0001, as determined by unpaired t-test. **(C)** Equal amounts of cell lysates were probed with antibodies for RpoA, IscR, YopE, and LcrF. T3SS cargo proteins secreted into culture supernatant and precipitated with TCA were probed with YopE antibody. All samples shown are from the same experiment, with each lysate split into two gels, one subsequently used for IscR and YopE probing and the other for LcrF probing. RpoA was blotted on each membrane as a loading control. One representative experiment out of three is shown.

oxygen, under these anaerobic fermentative conditions iron-limitation induced expression of *iscR* by 430-fold when compared to iron-replete conditions (Fig 4A). Similarly, expression of *lcrF* was induced 260-fold by iron-limitation. LcrF protein levels were also upregulated by iron limitation (Figs 4B and S4). Likewise, iron limitation upregulated expression of the LcrF target

# Aerobic conditions

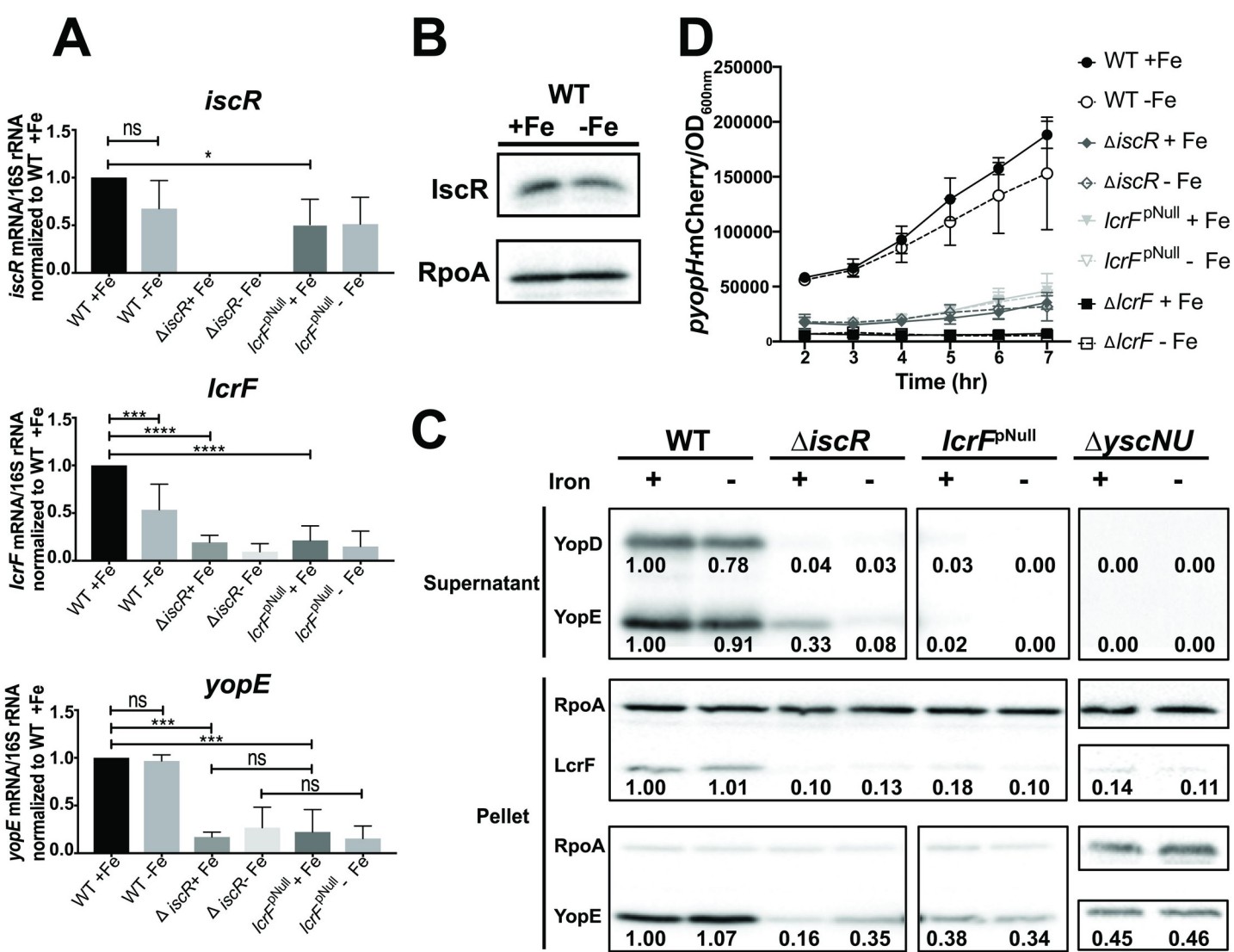

**Fig 3. Iron limitation does not affect IscR, LcrF, YopH, or YopE expression in the presence of oxygen.** M9/high glucose iron-limited aerobic cultures of *Y. pseudotuberculosis* were subsequently grown in iron-replete (+Fe) or iron-limited (-Fe) aerobic conditions and type III secretion induced by shifting to 37˚C for 4 hrs. (**A**) RNA was isolated and qPCR used to determine relative expression of *iscR*, *lcrF*, and *yopE* by normalizing to 16S rRNA levels. Data shown represent the average of four independent experiments. $^*p \leq 0.05$, $^{***}p \leq 0.001$, $^{****}p \leq 0.0001$, as determined by one-way ANOVA with Tukey post-test. (**B**) IscR and RpoA protein levels were determined by Western blot analysis. One representative experiment out of four is shown. (**C**) T3SS cargo proteins secreted into culture supernatant and precipitated with TCA were probed with antibodies for YopE and YopD. Cell lysates were probed with antibodies for RpoA, YopE, YopD, and LcrF. The $\Delta yscNU$ negative control strain does not express a functional T3SS. All samples shown are from the same experiment, with each lysate split into two gels, one subsequently used for LcrF probing and the other for YopE probing. RpoA was used as a loading control. One representative experiment out of four is shown. (**D**) *Y. pseudotuberculosis* P*yopH*-mCherry was used to assess *yopH* expression under M9/high glucose aerobic iron-replete (+Fe) or iron-limited (-Fe) conditions at 37˚C, in the genetic backgrounds indicated. mCherry fluorescence intensity normalized to OD$_{600}$ is shown, with T = 0 being the start of incubation at 37˚C. Data shown represent the average of three independent experiments. Densitometric quantification of the bands was performed using Image Lab software (Bio-Rad), setting the WT (+Fe) band to 1.00.

gene *yopE* by 1190-fold (p<0.0001; Fig 4A), and significantly induced Yop expression and secretion, as measured by detecting YopE and YopD protein levels in the pellet and supernatant (Figs 4B and S4). Additionally, shorter incubation under anaerobic conditions (4 hrs) led to similar results in type III secretion (S5 Fig), indicating that growth phase did not affect the

observed phenotype. These results suggest that iron depletion under anaerobic conditions leads to a shift from holo- to apo-IscR, de-repressing *iscR* expression and in turn inducing expression of LcrF, which then activates T3SS expression.

The *E. coli* T3SS was found to be potentiated by respiration [17]. We therefore examined *Yersinia* type III secretion in the presence of a terminal electron acceptor under anaerobic conditions. In M9 media lacking glucose but supplemented with mannitol as a carbon source and nitrate as an alternative terminal electron acceptor, we observed the same trend of increased IscR expression and T3SS activity under iron-limited conditions (Figs 5 and S6). Taken together, these data suggest that IscR expression is induced under aerobic conditions or by iron limitation under anaerobic conditions (during respiration or fermentation), and that this correlates with LcrF expression and ultimately T3SS activity.

### IscR control of LcrF is critical for type III secretion under anaerobic iron-limited conditions

In order to assess the role of IscR in the increased T3SS activity observed anaerobically in response to iron limitation, we examined gene expression and T3SS activity in the $\Delta iscR$ and $lcrF^{pNull}$ mutants. Under iron-limited anaerobic conditions, *lcrF* expression was reduced

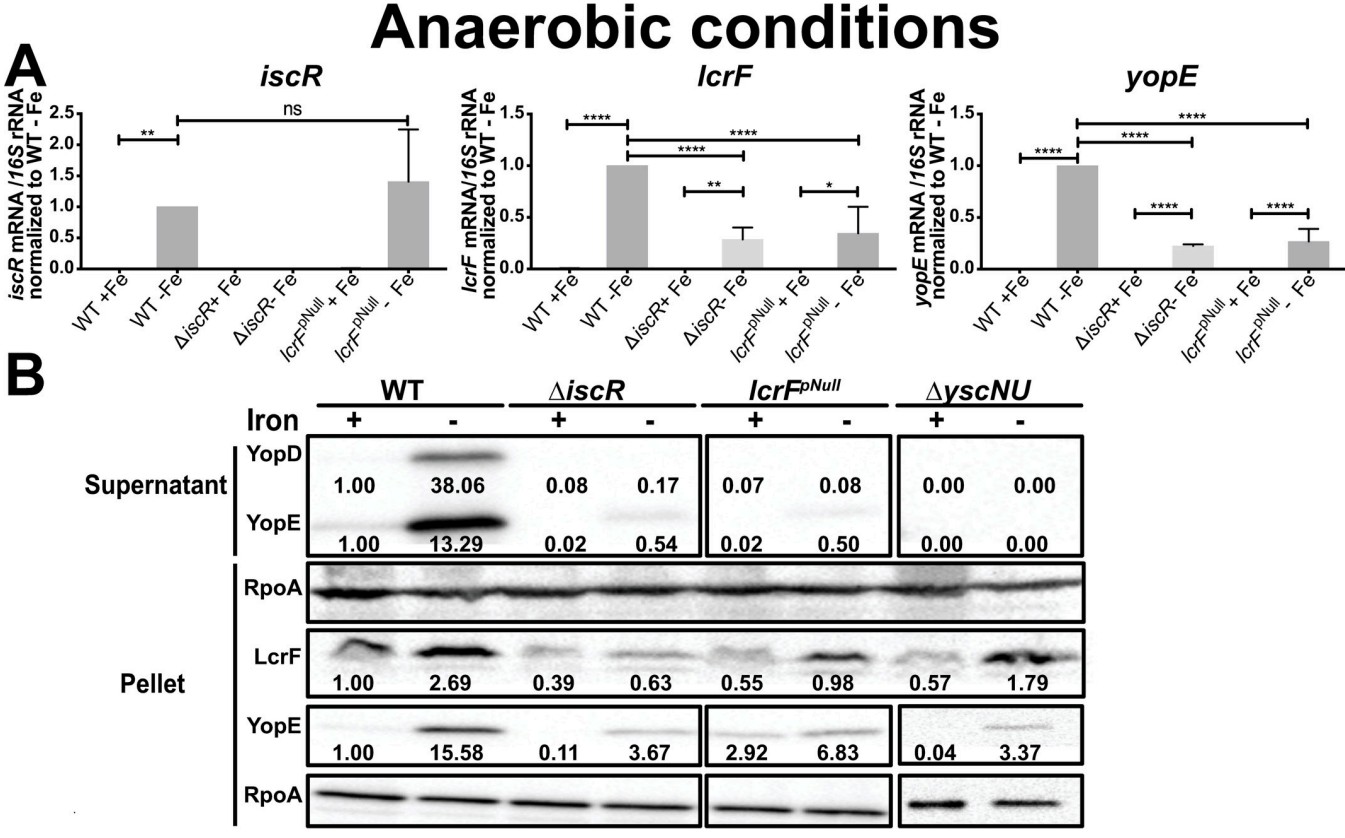

**Fig 4. Iron limitation induces *iscR*, *lcrF*, and *yopE* expression and type III secretion under fermentative conditions.** Iron-limited *Y. pseudotuberculosis* was shifted to iron-replete (+Fe) or iron-limited (-Fe) M9/high glucose media in the absence of oxygen and type III secretion induced at 37°C for 4 hrs. **(A)** RNA was isolated and qPCR used to determine relative expression of *iscR*, *lcrF*, and *yopE* by normalizing to 16S rRNA levels. Data shown represent the average of four independent experiments. $^*p \leq 0.05$, $^{**}p \leq 0.005$, $^{****}p \leq 0.0001$$^{****}p \leq 0.0001$, as determined by one-Way ANOVA with Tukey post-test. **(B)** T3SS cargo proteins secreted into culture supernatant and precipitated with TCA were probed with antibodies for YopE and YopD. Cell lysates were probed with antibodies for RpoA, YopE, and LcrF. All samples shown are from the same experiment, with each lysate split into two gels, one subsequently used for LcrF probing and the other for YopE probing. RpoA was used as a loading control. One representative experiment out of four is shown. Densitometric quantification of the bands was performed using Image Lab software (Bio-Rad), setting the WT (+Fe) band to 1.00.

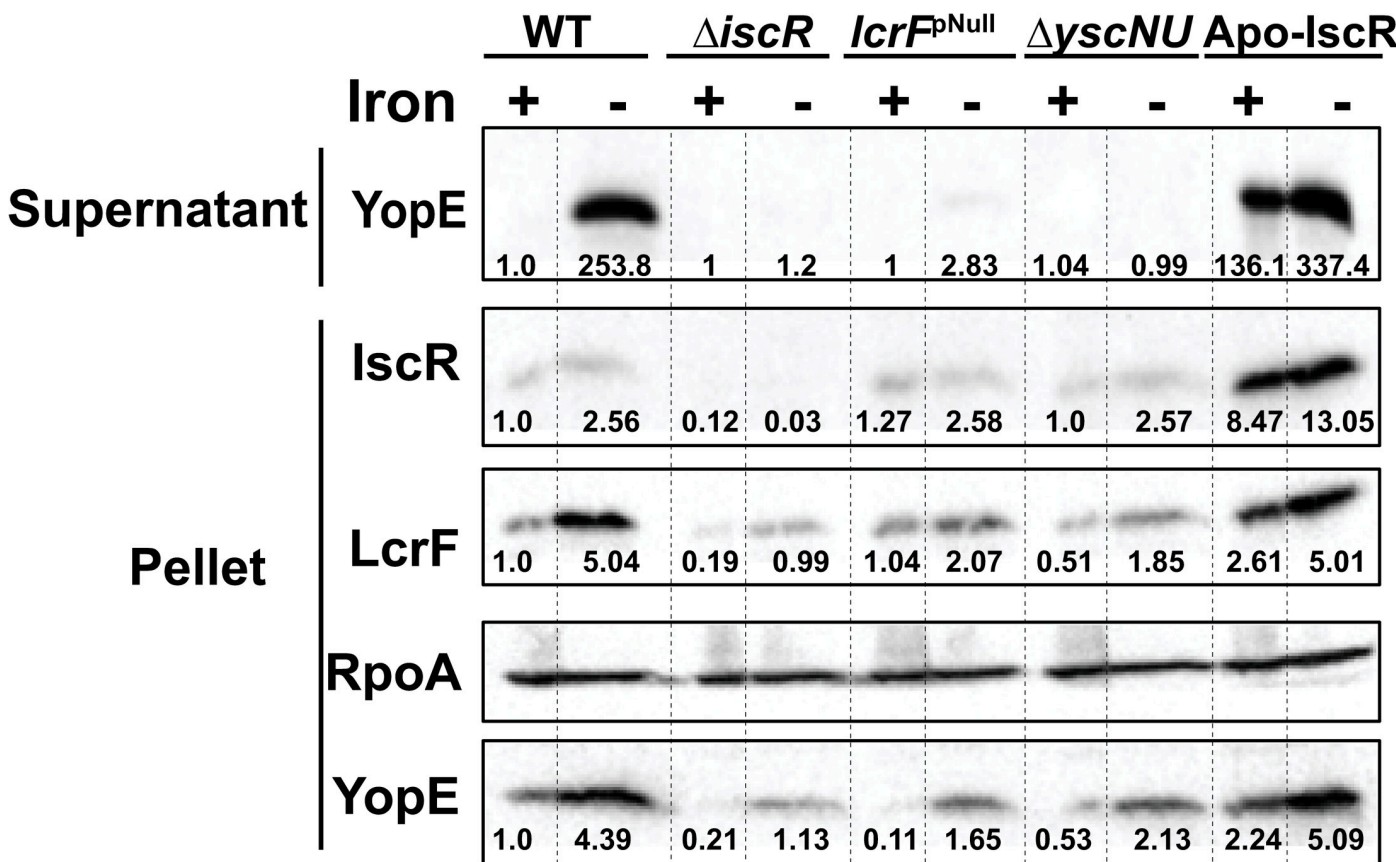

**Fig 5. Iron limitation induces *iscR*, *lcrF*, and *yopE* expression under anaerobic respiration conditions.** Iron-limited *Y. pseudotuberculosis* was shifted to iron-replete (+Fe) or iron-limited (-Fe) M9 media supplemented with sodium nitrate and mannitol in the absence of oxygen. Type III secretion was induced at 37˚C for 4 hrs. T3SS cargo proteins secreted into culture supernatant and precipitated with TCA were probed with antibodies for YopE. Cell lysates were probed with antibodies for RpoA, YopE, and LcrF. All samples shown are from the same experiment, with each lysate split into two gels, one subsequently used for LcrF and YopE probing and the other for IscR probing. RpoA was used as a loading control. One representative experiment out of three is shown. Dotted lines delimit the area of each well. Densitometric quantification of the bands was performed using Image Lab software (Bio-Rad), setting the WT (+Fe) band to 1.00.

~3-fold in both the Δ*iscR* or the *lcrF*^pNull mutant (Fig 4A) compared to WT, indicating a role of IscR in the upregulation of LcrF under these conditions. Indeed, there was no significant difference in *lcrF* expression between the Δ*iscR* and *lcrF*^pNull mutants in any of the conditions tested (Figs 3A and 4A), indicating that IscR increases *lcrF* transcription through direct binding to the *lcrF* promoter under both aerobic and anaerobic, iron-limiting conditions. Surprisingly, iron limitation induced a low level of expression of *lcrF* and *yopE* mRNA in the Δ*iscR* and *lcrF*^pNull mutants in the absence of oxygen, but this led to only trace amounts of secreted YopE and no detectable YopD in the supernatant of the mutants (Figs 4B, S4 and S5). These data suggest a minor IscR-independent mechanism of LcrF upregulation in response to iron limitation under anaerobic conditions. Taken all together, these results show that anaerobically-grown *Yersinia* increase T3SS expression in response to iron limitation through a predominantly IscR-dependent mechanism that requires IscR binding to the *lcrF* promoter.

## Ectopic induction of IscR under anaerobic conditions is sufficient to drive type III secretion

To test the hypothesis that upregulation of *iscR* expression is sufficient to induce expression of the T3SS, we used a *Y. pseudotuberculosis* apo-locked *iscR* mutant (Apo-IscR) in which the

three conserved cysteines are mutated to alanines (C93A, C98A, C104A), leading to loss of iron-sulfur cluster coordination. This mutant should constitutively express *iscR*, as it lacks holo-IscR to mediate *isc* promoter repression. We previously showed that under standard aerobic conditions, the Apo-IscR mutant induced expression of *lcrF* but was unable to secrete Yops due to a proton motive force defect [24]. As expected, the Apo-IscR mutant had high IscR levels under anaerobic conditions regardless of iron availability (Fig 5). Importantly, this correlated with high levels of LcrF and secreted YopE under both iron-replete and iron-limited conditions. Furthermore, in anaerobic cultures not starved for iron, we observed a 5-fold upregulation of IscR protein levels, a 6-fold increase in LcrF, and a 7-fold increase in YopE in the Apo-IscR mutant cell pellet compared to WT (Fig 6). Surprisingly, we also observed secreted YopE in the Apo-IscR mutant supernatant (Figs 5 and 6), indicating that either the Apo-IscR mutant does not have a proton motive force defect under anaerobic conditions or that the proton motive force is not required for type III secretion in the absence of oxygen. We observed a small amount of RpoA in the supernatant of the Apo-IscR mutant (Fig 6), indicative of cell lysis in this strain under anaerobic conditions. Therefore, it is possible that some of the YopE detected in the supernatant of the Apo-IscR strain is a result of lysis rather than type III secretion. However, the 5-fold greater YopE levels in the cell pellet of the Apo-IscR mutant compared to WT (Fig 6) corroborates the hypothesis that this mutant is undergoing type III secretion, as active secretion feeds back on T3SS gene transcript levels through a positive feedback loop [31]. These results demonstrate that increasing IscR levels is sufficient to activate T3SS expression under anaerobic conditions.

## Proper regulation of the T3SS by IscR is necessary for disseminated infection

To examine the impact of IscR-mediated regulation of *lcrF* on infection, we performed oral inoculation of mice using a bread feeding model. Consistent with our previous data using an oral gavage model [24], the Δ*iscR* mutant displayed decreased colonization in the Peyer's patches (PP, p = 0.0396) and a trend toward decreased colonization of the mesenteric lymph nodes (MLN) that was not statistically significant (p = 0.0962). This mutant also displayed a 200-fold defect in bacterial burden in the spleen and 460-fold in the liver (p<0.0001) (Fig 7). Although the *lcrF*^pNull^ mutant had a trend of lower PP and MLN colonization compared to WT (5-fold and 3-fold, respectively), this was not statistically significant (p = 0.1625 and p = 0.0997, respectively). Importantly, the *lcrF*^pNull^ mutant exhibited severe defects in disseminated infection (100-fold in the spleen and 90-fold in the liver; p = 0.0007 and p = 0.0005, respectively). Interestingly, the Δ*iscR* mutant was defective at colonizing the small intestine, while the *lcrF*^pNull^ mutant was comparable to WT. These results suggest that IscR binding to the *lcrF* promoter is critical for *Yersinia* disseminated infection while IscR itself is also important for *Yersinia* colonization of the intestine and Peyer's patches.

## IscR is required for type III secretion in *Yersinia pestis*

The region containing the IscR binding site upstream of *lcrF* is 100% identical between *Y. pseudotuberculosis* and *Y. pestis* [32]. Therefore, we sought to determine if T3SS activity in *Y. pestis* also required IscR despite the difference in life cycles carried out by enteropathogenic *Yersinia* compared to *Y. pestis*. To assess T3SS activity *in vitro* we incubated WT and Δ*iscR* strains of pgm⁻ *Y. pestis* in BHI media at 37˚C in the presence of oxygen. Calcium depletion serves as a host-cell independent trigger of type III secretion *in vitro* [33]. Importantly, deletion of *iscR* led to significantly decreased YopE secretion (Fig 8), indicating that IscR is a critical regulator of the T3SS in both *Y. pseudotuberculosis* and *Y. pestis*.

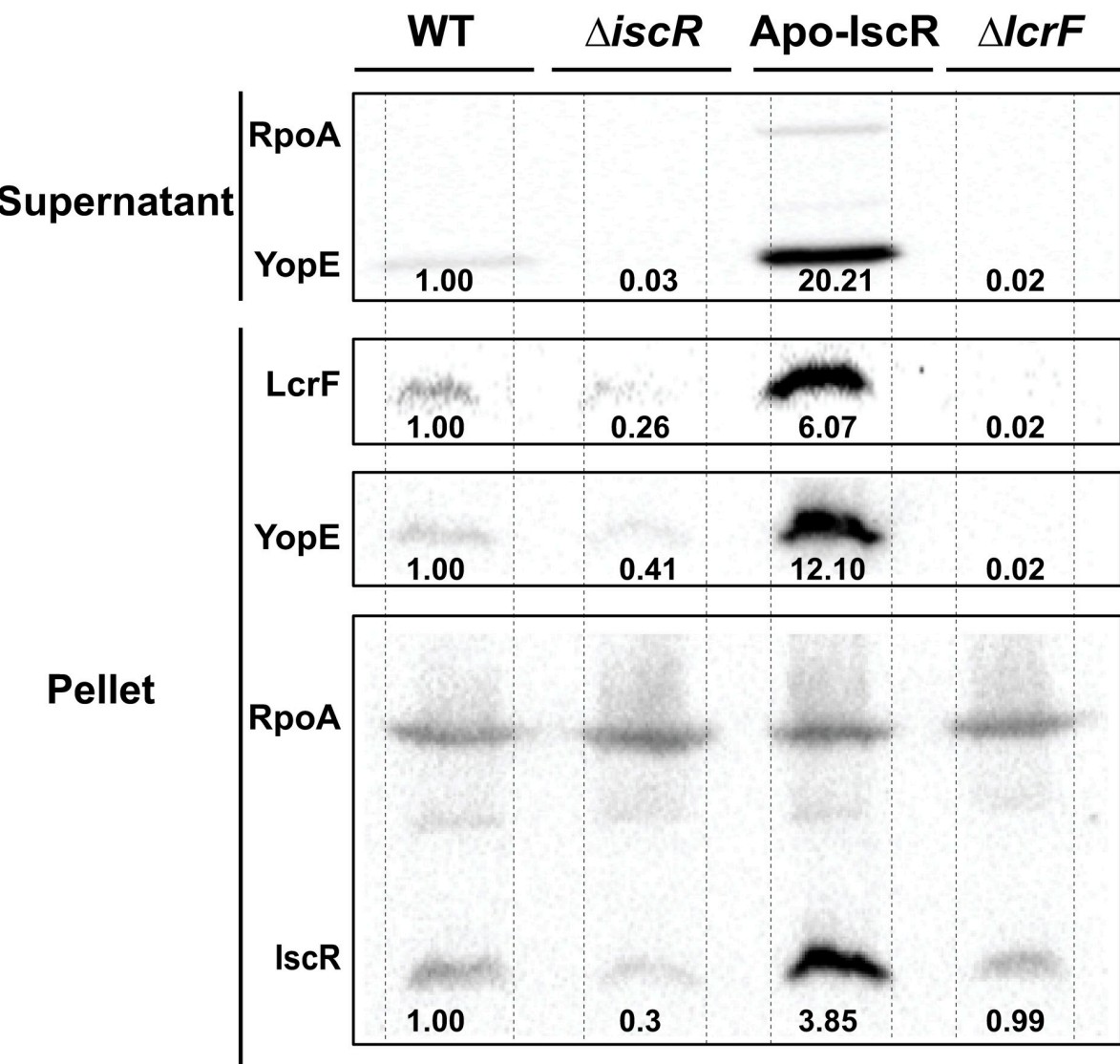

**Fig 6. Ectopic expression of IscR is sufficient to rescue T3SS expression under anaerobic, iron-replete conditions.** *Y. pseudotuberculosis* was grown in iron-replete M9/high glucose and in the absence of oxygen and type III secretion induced at 37˚C for 4 hrs, as shown in Fig 2. T3SS cargo proteins secreted into culture supernatant and precipitated with TCA were probed with an antibody for YopE. Cell lysates were probed with antibodies for RpoA, IscR, YopE, and LcrF. All samples shown are from the same experiment, with each lysate split into two gels, one subsequently used for LcrF and YopE probing and the other for IscR probing. RpoA was used as a loading control. One representative experiment out of three is shown. Dotted lines delimit the area of each well. Densitometric quantification of the bands was performed using Image Lab software (Bio-Rad), setting the WT band to 1.00.

## Discussion

A recent study suggested that anaerobiosis may repress the Ysc T3SS expression *in vitro* [18], although the mechanism underlying this regulation was unknown. Here we show that the *Y. pseudotuberculosis* global transcription factor IscR regulates the Ysc T3SS in response to oxygen and iron availability and is dependent upon IscR binding to the promoter of the T3SS master regulator *lcrF*. Specifically, we show that conditions that increase Fe-S cluster demand, such as aerobic conditions or iron limitation, induce IscR expression and, in turn, IscR upregulates LcrF and subsequently the T3SS. IscR regulation of LcrF and the T3SS is critical for *Y.*

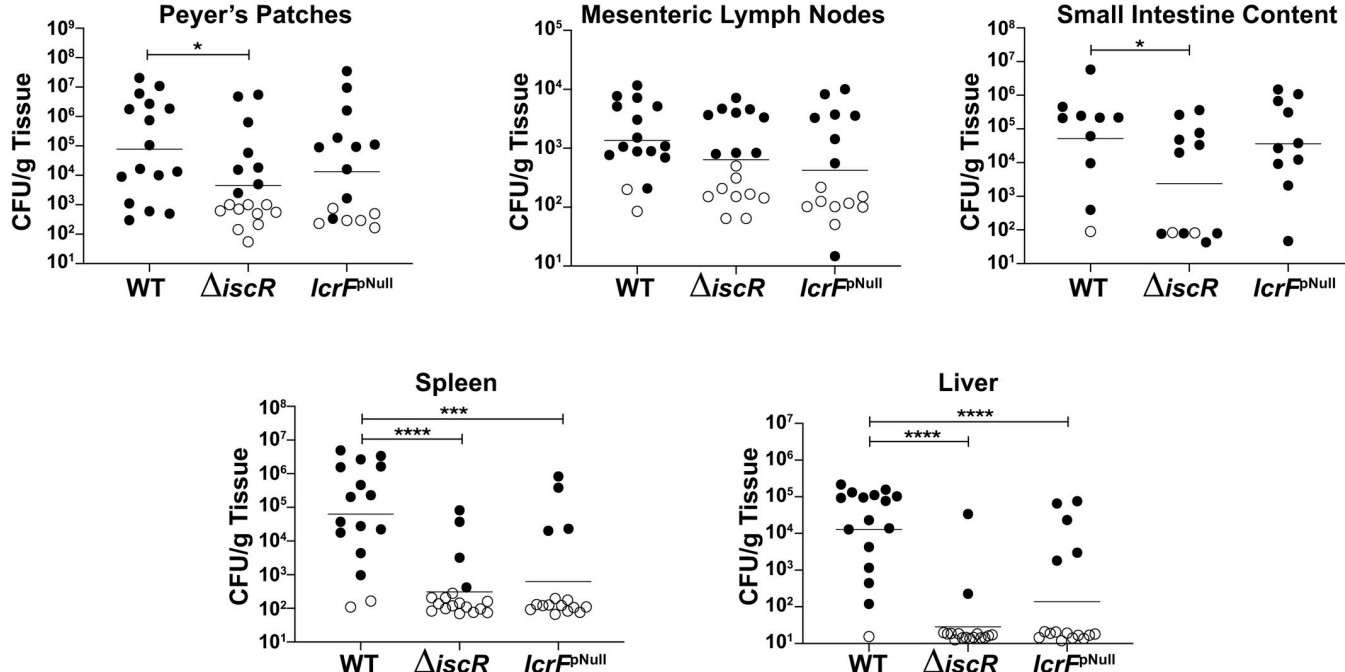

**Fig 7. Proper regulation of *lcrF* by IscR is necessary for disseminated infection.** Mice were infected with ~2x10⁸ *Y. pseudotuberculosis* using a bread feeding model, and organs and intestinal contents harvested 5 days post-inoculation. CFUs were normalized to the weight of the organ in grams. Each symbol represents one animal. Unfilled symbols are set at the limit of detection for each individual organ based on weight, and represent CFU that were below this limit. Graphs show data combined from five independent experiments. **p<0.01, *** p<0.001, ****p <0.0001, as determined by an unpaired Mann-Whitney rank sum test. Dashes represent geometric means.

*pseudotuberculosis* disseminated infection and is conserved between *Y. pseudotuberculosis* and *Y. pestis*. These data provide a mechanism by which oxygen and iron control expression of the *Yersinia* T3SS.

LcrF is an AraC-like DNA binding transcriptional regulator of the T3SS in *Yersinia*, controlling expression of the majority of T3SS structural and regulatory proteins, as well as the Yop effector proteins [34, 35]. We previously demonstrated that IscR is essential for LcrF expression and binds to a type II motif just upstream of the *lcrF* promoter [24]. In addition, LcrF is known to be tightly regulated both at the transcriptional and translational level in response to temperature and calcium. LcrF expression is upregulated at 37°C, mammalian

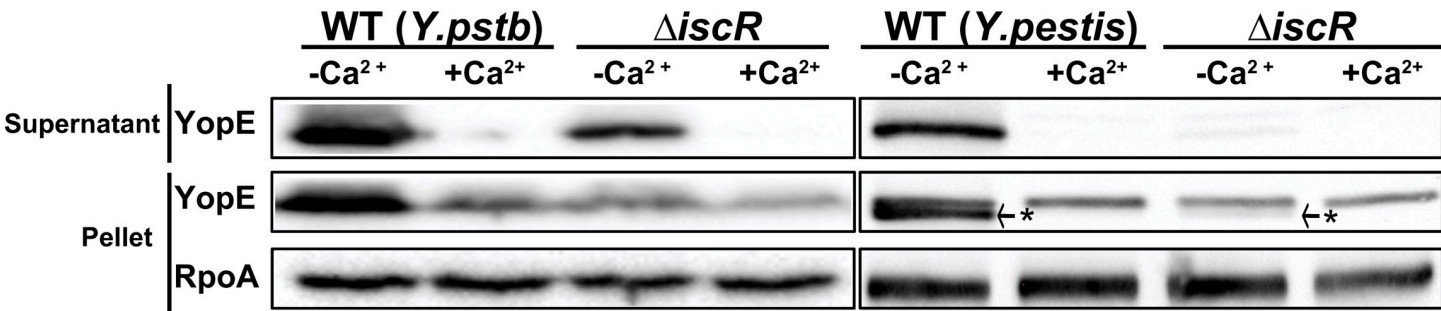

**Fig 8. IscR is critical for type III secretion in *Y. pestis*.** WT and ΔiscR *Y. pseudotuberculosis (pstb)* and *Y. pestis* pgm- were grown in BHI low calcium or high calcium media at 37°C. Proteins in the bacterial culture supernatant were precipitated with TCA and type III secretion probed using an anti-YopE antibody. Results are representative of three independent experiments. * YopE specific band. Doublet bands for YopE have been previously reported [58].

host body temperature, and upon calcium depletion, which is thought to mimic contact with host cells [33, 35–38]. LcrF thermal control is mediated by host temperature dependent degradation of the negative regulator YmoA to activate *lcrF* transcription at mammalian body temperature and an RNA thermometer that represses LcrF translation at environmental temperatures [32, 35, 38, 39]. A recent study showed that low calcium or host cell contact lead to secretion of a T3SS cargo protein and regulator, YopD, that ultimately affects *lcrF* mRNA stability [40]. Stabilization of *lcrF* and other mRNAs following secretion of YopD may therefore amplify differences in transcription of *lcrF* when comparing bacterial strains with differences in their ability to secrete YopD. However, in this study we demonstrated that, in addition to temperature and calcium, iron availability and oxygen tension control *Yersinia* T3SS expression through a mechanism important for pathogenesis. This regulatory pathway was largely dependent on IscR binding to the *lcrF* promoter.

Previous studies of IscR in *E. coli* provide a framework for explaining the oxygen and iron dependent regulation invoked by *Yersinia* IscR observed in our studies. *iscR* is the first gene in the operon encoding the Isc Fe-S cluster biogenesis pathway. Under anaerobic conditions IscR maximally represses this operon, including its own gene, through a negative feedback loop that requires binding of holo-IscR to two type I motifs upstream of the *isc* promoter, leading to low levels of IscR [23, 25, 30]. Indeed, Δ*iscR Y. pseudotuberculosis* displays enhanced expression of the *isc* operon, indicating that this negative regulation is also present in *Yersinia* [24]. This suggests that under conditions where Fe-S cluster turnover is low, such as in the absence of oxygen, holo-IscR should repress *isc* transcription in *Yersinia*, maintaining low overall IscR levels. Conversely, under conditions where Fe-S cluster demand is high, such as high oxygen tension, the majority of IscR should be clusterless, de-repressing IscR [30]. In this study we demonstrated that IscR expression is induced under aerobic conditions when compared to anaerobic conditions. However, under anaerobic conditions IscR can be induced by depletion of iron, presumably as a result of holo-IscR depletion leading to derepression of *isc* transcription and ultimately an increase in apo-IscR levels. As shown here, changes in the amount of IscR cause downstream changes in the expression of the T3SS through direct transcriptional control of the major regulator LcrF.

Our data demonstrate that loss of IscR leads to virulence defects distinct from those associated specifically with loss of IscR control of *lcrF* transcription. Upon ingestion, *Y. pseudotuberculosis* is exposed to the mucosal environment of the gastrointestinal tract, and, once in the lumen of the small intestine, it binds to the apical surface of microfold (M) cells to subsequently translocate into the Peyer's patches, where it replicates extracellularly and colonizes mesenteric lymph nodes (MLNs) [41, 42]. The *Y. pseudotuberculosis* T3SS contributes to early colonization of the Peyer's patches and MLNs two days post-inoculation, but is not essential for colonization of these tissues by day five [43], consistent with the lack of a statistically significant defect for the *Y. pseudotuberculosis lcrF*[pNull] mutant in colonization of these tissues at five days post-inoculation. However, colonization of the Peyer's patches and MLN is not required for *Y. pseudotuberculosis* disseminated infection [44]. Instead, *Y. pseudotuberculosis* intestinal replication is required for subsequent successful infection of the spleen and liver through an unknown pathway [44, 45]. Interestingly, a recent publication postulated the existence of a "gut vascular barrier" that only allows passage of small antigens under 70 kDa from the gut lumen, across the blood endothelial cells, and to the bloodstream. However, infection of *S. enterica* serovar Typhimurium disrupts this barrier, allowing bacteria to cross into the bloodstream and disseminate to the spleen and liver [46]. Notably, our data demonstrated that the *Y. pseudotuberculosis* Δ*iscR* mutant was defective in colonization of the small intestinal lumen, Peyer's patches, spleen, and liver. Therefore, it is possible the Δ*iscR* mutant may be unable to initiate the undefined colonization pathway from the intestinal lumen to the spleen and liver

in addition to being defective for growth in hepatosplenic tissue. In contrast, the $lcrF^{pNull}$ mutant did not display an intestinal colonization defect but failed to cause disseminated infection, as significantly lower hepatosplenic burden was observed compared to WT. These data support the hypothesis that IscR regulates multiple factors required for colonization of mammalian tissues, consistent with data showing that IscR regulates various virulence factors in other bacteria [27, 47]. In contrast, IscR regulation of LcrF and the T3SS specifically disrupts the ability to establish successful disseminated infection.

The need for *Yersinia* to sense iron and oxygen availability to control T3SS expression may reflect a requirement to repress T3SS expression in the intestinal lumen and induce T3SS expression in deeper tissues. Previous studies suggest that *Yersinia* encounter differing oxygen tension and iron availability within the mammalian host. In the intestinal lumen, iron and heme uptake systems are downregulated in *Y. enterocolitica*, suggesting that iron is readily available [48]. The intestinal lumen is also an anaerobic environment, as a result of the concerted action of obligate anaerobes [13]. In contrast, previous data suggests that three days post-inoculation in lymph tissue, *Yersinia* encounter an anaerobic, iron-limited environment [49, 50], conditions we have shown *in vitro* to induce *Yersinia* type III secretion through IscR. For example, during *Y. pseudotuberculosis* infection, nutritional immunity factors are among the highest expressed genes in infected lymphatic tissue. Specifically, the iron sequestration molecules haptoglobin and lipocalin-2 were >100-fold induced during infection [50]. In addition, there is a concomitant upregulation of *Yersinia* iron uptake systems, as well as genes involved in anaerobic growth. As IscR control of the T3SS is critical for colonization of the spleen and liver, it is likely that conditions in those tissues are also conducive to IscR and therefore LcrF expression. Indeed, Davis and colleagues examined *Y. pseudotuberculosis* gene expression within microcolonies in the spleen [51]. While T3SS gene expression was detected within the entire splenic microcolony, enhanced expression was detected in bacteria localized at the microcolony periphery, i.e.-those bacteria in contact with host cells. It is possible that this T3SS expression in the microcolony periphery reflects host cell contact-dependent increase in type III secretion, leading to stabilization of *lcrF* and other T3SS genes [31]. This contact-dependent increase in T3SS gene expression is then an additional layer of regulation once IscR levels are high enough in deeper tissues to induce sufficient *lcrF* mRNA production.

Deletion of *iscR* in *Y. pestis* led to a severe defect in type III secretion. IscR and the IscR binding site within the *lcrF* promoter is completely conserved between *Y. pseudotuberculosis* and *Y. pestis*, yet *Y. pestis* is transmitted through flea or aerosol infection routes and not through the intestine. Considering that *Y. pestis* split from *Y. pseudotuberculosis* between 1,500–6,400 years ago [52], it is possible that IscR induction of the T3SS was retained by *Y. pestis* as a mechanism to promote sufficient T3SS expression for growth in lymph nodes, lung, spleen, and liver. Indeed, analysis of *Y. pestis* gene expression in the rat bubo suggests that *Yersinia* encounters anaerobic, iron-limited conditions that we predict would be conducive to elevated IscR and therefore LcrF expression. Furthermore, the aerobic conditions in the lung should also lead to high IscR levels.

Recently, a study in *Salmonella enterica* serovar Typhimurium revealed that IscR regulates the SPI-1 T3SS in response to iron and oxygen availability. Specifically, a type II DNA binding motif was identified upstream of the *hilD* gene, the master regulator of the SPI-1 T3SS. These data suggested that IscR represses *hilD* expression during iron depletion or aerobic conditions when IscR is upregulated, but *hilD* expression is derepressed under anaerobic conditions when IscR levels are decreased. This would allow *S. enterica* to upregulate SPI-1 T3SS expression in the anaerobic environment of the small intestine in order for *Salmonella* to invade the intestinal epithelium. Once *Salmonella* gains entry into the iron-depleted environment of the host cell, *Salmonella* no longer requires the SPI-1 T3SS. Iron limitation in deeper tissues would

upregulate IscR expression and this should in turn repress expression of *hilD* and, subsequently, the SPI-1 T3SS [22]. Interestingly, the way in which IscR controls the *Salmonella* T3SS is distinct from IscR-induced T3SS expression in *Y. pseudotuberculosis*, which requires its T3SS once the bacteria cross the intestinal epithelium and, presumably, encounter increased oxygen tension or iron limitation under hypoxic conditions. It is noteworthy that two bacterial pathogens have coopted IscR, which is found in numerous pathogenic and non-pathogenic bacteria, to regulate T3SS genes encoded on horizontally-acquired genetic elements in a way that promotes their unique life cycles.

## Materials and methods

All animal use procedures were in strict accordance with the NIH Guide for the Care and Use of Laboratory Animals and were approved by the UC Santa Cruz Institutional Animal Care and Use Committee (protocol Stona1903). Animals were euthanized with an overdose of isoflurane followed by cervical dislocation.

### Bacterial strains and growth conditions

The strains used in this study are listed in Table 1. For standard *Y. pseudotuberculosis* growth and T3SS induction (Fig 1C), bacteria were grown overnight in 2xYT media, subcultured in 2xYT plus 20 mM sodium oxalate (to chelate calcium and induce type III secretion) and 20 mM $MgCl_2$ to an optical density at 600 nm ($OD_{600}$) of 0.2, grown at 26˚C for 1.5 hrs followed by 37˚C for another 1.5 hrs. For growing *Yersinia* under various iron and oxygen condition, casamino acid-supplemented M9 media, referred to as M9 below, was used [53].

Growth of cultures to vary oxygen tension (Figs 2 and 6) was achieved by first diluting 26˚C overnight aerobic cultures of *Y. pseudotuberculosis* to an $OD_{600}$ of 0.1 in fresh M9 minimal media supplemented with 0.9% glucose (referred to as M9/high glucose) to maximize growth rate and energy production under anaerobic conditions, and incubating for 12 hrs under either aerobic or anaerobic conditions at 26˚C. Anaerobic cultures were then shifted to 37˚C and incubated for 4 hrs in a vinyl anaerobic chamber with a gas mix of 90%N, 5%$CO_2$, 5% $H_2$ (Coy Laboratory Products, Inc). Aerobic cultures were instead diluted to an $OD_{600}$ of 0.2, grown with agitation for 2 hrs at 26˚C, and then shifted to 37˚C for 4 hrs. Where indicated, to test for effect of anaerobic respiration, M9 media was supplemented with 40 mM sodium nitrate as an alternative electron acceptor and 40 mM mannitol as a carbon source instead of glucose.

Growth of cultures to vary iron availability (Figs 4 and 5) was achieved by first growing *Y. pseudotuberculosis* aerobically in M9/high glucose media treated with Chelex 100 resin to remove all traces of iron in acid-washed glassware, as previously described [29]. Specifically,

**Table 1. Strains used in this study.**

| Strain | Background | Mutation(s) | Reference |
|---|---|---|---|
| WT | IP2666 | Naturally lacks full-length YopT | [59] |
| Δ*iscR* | IP2666 | Δ*iscR* | [24] |
| Δ*yscNU* | IP2666 | Δ*yscNU* | [43] |
| *lcrF*$^{pNull}$ | IP2666 | Point mutations in IscR binding site on p*lcrF* | This study |
| Δ*iscR* | *Y. pestis* KIM D27 pgm- | Lacks pgm locus, Δ*iscR* | This study |
| WT | *Y. pestis* KIM D27 pgm- | Lacks pgm locus | [60] |
| p*yopH*-mCherry | IP2666 | pMMB67EH expressing mCherry under the control of the YopH promoter | [61] |

iron-replete overnight cultures (M9+6.58 μM $FeSO_4$) grown at 26˚C aerobically were diluted to an $OD_{600}$ of 0.1 into Chelex-treated M9/high glucose media and grown for 8 hrs at 26˚C aerobically with agitation. Cultures were then diluted a second time to $OD_{600}$ 0.1 in fresh Chelex-treated M9/high glucose and grown for 12 hrs at 26˚C with agitation. Subsequently, cultures were then diluted a third time to $OD_{600}$ 0.2 in M9/high glucose supplemented with 6.58 μM $FeSO_4$ (iron-replete) or with 0.0658 μM $FeSO_4$ (iron limitation), grown for 2 hrs at 26˚C with agitation, and then shifted to 37˚C for 4 hrs with agitation to induce type III secretion. For anaerobic cultures, the cultures were instead diluted a second time to $OD_{600}$ 0.1 in M9/high glucose supplemented with 6.58 μM $FeSO_4$ (iron-replete,) or with 0.0658 μM $FeSO_4$ (iron limitation), and transferred to a vinyl anaerobic chamber where they were grown at 26˚C for 12 hrs (most experiments) or 4 hrs (experiment shown in S5 Fig). Rezasurin redox indicator was used to monitor oxygen levels in the anaerobic chamber. By 4 hrs after introduction of cultures to the anaerobic chamber, the dye color changed from pink to clear, indicating an absence of residual oxygen. Cultures were then shifted to 37˚C for another 4 hrs to induce type III secretion.

*Y. pestis* KIM D27 pgm$^-$ was grown in BHI (brain heart infusion) at 26˚C with agitation overnight. Cultures were diluted into low-calcium medium (BHI plus 20 mM sodium oxalate and 20 mM $MgCl_2$) or high-calcium medium (BHI plus 2.5 mM $CaCl_2$) to $OD_{600}$ of 0.1 and grown for 3 hrs at 26˚C with agitation, followed by 2 hrs at 37˚C with agitation to induce type III secretion.

## Type III secretion assays

Visualization of T3SS cargo secreted in broth culture was performed as previously described [54]. Briefly, 5 mL of each culture were pelleted at 13,200 rpm for 15 min at room temperature. Supernatants were removed, and proteins precipitated by addition of trichloroacetic acid (TCA) at a final concentration of 10%. Samples were incubated on ice overnight and pelleted at 13,200 rpm for 15 min at 4˚C. Resulting pellets were washed twice with ice-cold 100% acetone and subsequently resuspended in final sample buffer (FSB) containing 20% dithiothreitol (DTT). Samples were boiled for 5 min prior to running on a 12.5% SDS-PAGE gel.

## Western blot analysis

In parallel, after the supernatant was collected, bacterial pellets were resuspended in FSB plus 20% DTT. Pellet samples were boiled for 15 min. At the time of loading, samples were normalized to the same number of cells. Cytosolic and secreted protein samples were run on a 12.5% SDS-PAGE gel and transferred to a blotting membrane (Immobilon-P) with a wet mini trans-blot cell (Bio-Rad). Blots were blocked for an hour in Tris-buffered saline with Tween 20 and 5% skim milk, and probed with the goat anti-YopE antibodies (Santa Cruz Biotechnology), rabbit anti-YopD (gift from Alison Davis and Joan Mecsas), rabbit anti-RpoA (gift from Melanie Marketon), rabbit anti-LcrF [35], rabbit anti-IscR [25], and horseradish peroxidase-conjugated secondary antibodies (Santa Cruz Biotech). Following visualization, quantification of the bands was performed with Image Lab software (Bio-Rad).

## Construction of *yersinia* mutant strains

For the *Y. pseudotuberculosis lcrF*$^{pNull}$ mutant strain, the *lcrF* promoter region was amplified by PCR using 5'/3' *lcrF*$^{pNull}$ which contained the mutations in the IscR binding site (see Table 2, underlined nucleotides). Amplified PCR fragments were cloned into a *Bam*HI- and SacI-digested pSR47s suicide plasmid (λpir-dependent replicon, kanamycin$^R$, *sacB* gene conferring sucrose sensitivity) using the NEBuilder HiFi DNA Assembly kit (New England

**Table 2. Primers used in this study.**

| Name | Name Sequence Reference | Reference |
|------|------------------------|-----------|
| FqyopE | CCATAAACCGGTGGTGAC | [62] |
| RqyopE | CTTGGCATTGAGTGATACTG | [62] |
| Fq16s | AGCCAGCGGACCACATAAAG | [56] |
| Rq16s | AGTTGCAGACTCCAATCCGG | [56] |
| FqlcrF | GGAGTGATTTTCCGTCAGTA | [24] |
| RqlcrF | CTCCATAAATTTTTGCAACC | [24] |
| FqiscR | CAGGGCGGAAATCGCTGCCT | This study |
| RqiscR | ATTAGCCGTTGCGGCGCCTAT | This study |
| 3' *lcrF*[pNull] | TGTATAAATCCTTTGAAATCGCATCATATATTCCTAATAT | This study |
| 5' *lcrF*[pNull] | GATTTCAAAGGATTTATACAGTATGGTAATTGTATTTCT | This study |

Biolabs, Inc) [55, 56]. Recombinant plasmids were transformed into *E. coli* S17-1 λpir competent cells and later introduced into *Y. pseudotuberculosis* IP2666 via conjugation. The resulting Kan[R], irgansan[R] (*Yersinia* selective antibiotic) integrants were grown in the absence of antibiotics and plated on sucrose-containing media to select for clones that had lost *sacB* (and by inference, the linked plasmid DNA). Kan[S], sucrose[R], congo red-positive colonies were screened by PCR and sequencing. The *Y. pestis* Δ*iscR* strain was generated using a pSR47S:Δ*iscR* suicide plasmid as described previously for *Y. pseudotuberculosis* [24], except using BHI plates instead of LB plates. The pSR47S:Δ*iscR* suicide plasmid was used because the *iscR* gene and the 700 bp regions upstream and downstream encoded on the plasmid are 100% identical between *Y. pestis* and *Y. pseudotuberculosis*.

## Mouse model

15–18 h prior to infection, food was withheld from eleven to twelve week old 129S1/SvImJ mice, but water provided ad libitum. Mice were then inoculated with $2 \times 10^8$ by bread feeding [57]. Briefly, *Y. pseudotuberculosis* cultures were grown in LB media overnight at 26˚C. Cultures were diluted to the appropriate $OD_{600}$ to obtain $2 \times 10^8$ bacteria per mouse. Dilutions were spun down and pellets resuspended in a mixture of 2 μl of 1X PBS and 3 μl melted butter, and this mixture was then pipetted onto a small piece of bread. Mice were given a piece of *Yersinia*-soaked bread and provided food and water *at libitum*. Five days post-inoculation, mice were euthanized and Peyer's patches, mesenteric lymph nodes, spleens, and livers were isolated and homogenized for 30 s in PBS followed by serial dilution and plating on LB supplemented with 1 mg/mL irgasan for CFU determination. Significant difference in colonization was determined by an unpaired Mann-Whitney rank sum test to compare each mutant strain against WT.

## Electrophoretic mobility shift assays (EMSAs)

*E. coli* IscR-C92A protein that lacks the [2Fe-2S] cluster was isolated as previously described [25, 30] and subsequently used in electrophoretic mobility shift assays (EMSAs) because this apo- form of IscR binds exclusively to type II sites. DNA fragments containing the wild-type *Y. pseudotuberculosis lcrF* promoter (-206 to +12 bp relative to the +1 transcription start site), or its *lcrF*[pNull] variant in which the IscR binding site is disrupted, were isolated from pPK12778 and pPK12779, respectively, after digestion with *Hind*III and *Bam*HI, and EMSAs were carried out with purified IscR-C92A as previously described [29]. Cut vector backbone is also present in the reaction and serves as a source of nonspecific DNA. After incubation at 37˚C for 30

min, samples were loaded onto a non-denaturing 6% polyacrylamide gel in 0.5× Tris-borate-EDTA buffer and run at 100 V for 4.0 hrs at room temperature. The gel was stained with SYBR Green EMSA nucleic acid gel stain (Molecular Probes) and visualized using a Typhoon FLA 900 imager (GE).

## YopH-mCherry transcriptional reporter assay

Overnight bacterial cultures grown in M9 media were back diluted and iron starved as described above, and then shifted to 37˚C to induce type III secretion. Starting at 2 hrs after shift, 1.5 mL of cells were spun down at 3.5 rcf for five minutes, resuspended in 200μl of 1x PBS, and mCherry fluorescence measured in black bottom 96 well plates (Costar). In parallel, optical density was measured in clear bottom 96 well plates (Costar). Fluorescence and optical density were both measured using a Perkin Elmer Victor X3 plate reader.

## RNA isolation and quantitative PCR (qPCR) analysis

A total of 3 mL of culture from each condition were pelleted by centrifugation for 5 minutes at 4,000 rpm. The supernatant was removed, and pellets were resuspended in 500 μL of media and treated with 1 mL Bacterial RNA Protect Reagent (Qiagen) according to the manufacturer's protocol. Total RNA was isolated using the RNeasy Mini Kit (Qiagen) per the manufacturer's protocol. After harvesting total RNA, genomic DNA was removed via the TURBO-DNA-free kit (Life Technologies/Thermo Fisher). cDNA was generated for each sample by using the M-MLV Reverse Transcriptase (Invitrogen) according to the manufacturer's instructions, as previously described [24]. Each 20 μl qPCR assay contained 5 μl of 1:10 diluted cDNA sample, 10 μl of Power CYBR Green PCR master mix (Thermo Fisher Scientific), and primers (Table 2) with optimized concentrations. The expression levels of each target gene were normalized to that of 16S rRNA present in each sample and calculated by the ΔΔCt method. Three independent biological replicates were analyzed for each condition.

## Supporting information

**S1 Fig. *Y. pseudotuberculosis lcrF^{pNull}* mutant does not display a growth defect.** Overnight bacterial cultures were subcultured into fresh M9 media containing iron and grown under aerobic condtions for 9 hrs at either 26˚C or 37˚C. $OD_{600}$ was measured every hour. Note that the pYV⁻ mutant lacking all T3SS genes does not undergo the normal growth arrest seen in wild-type *Yersinia* following T3SS induction at 37˚C [63].
(TIF)

**S2 Fig. Iron availability does not impact type III secretion under aerobic conditions.** Two additional independent replicates of the results shown in Fig 3 are provided. Iron-limited *Y. pseudotuberculosis* were induced for the T3SS under aerobic iron-replete (+Fe) or iron-limited (-Fe) conditions as described in Fig 3 and assayed for proteins by Western blots. Top panels, supernatant. Bottom panels, cell pellet.
(TIF)

**S3 Fig. *Y. pseudotuberculosis lcrF^{pNull}* displays similar growth rate to the WT and Δ*iscR* strains under iron-limited, anaerobic conditions.** *Y. pseudotuberculosis* strains were iron starved under anaerobic conditions and $OD_{600}$ was measured every hour after shifting to 37˚C. Data shown represent the average of three independent experiments.
(TIF)

**S4 Fig. Type III secretion is induced by iron limitation following 12 hours of anaerobic growth.** *Y. pseudotuberculosis* was iron starved and grown for 12 hours in the absence of oxygen prior to inducing the T3SS by shifting to 37˚C, as in Fig 4. Two additional independent replicates are shown. Top panels, supernatant. Bottom panels, cell pellet.
(TIF)

**S5 Fig. Type III secretion is induced by iron limitation following only 4 hrs of anaerobic growth.** *Y. pseudotuberculosis* was iron starved and grown for only four hours in the absence of oxygen prior to inducing the T3SS by shifting to 37˚C. Secreted proteins were precipitated with TCA and analyzed by Western blot. Two independent experiments are shown.
(TIF)

**S6 Fig. Iron depletion induces IscR, LcrF, and YopE expression during anaerobic respiration.** Iron starved *Y. pseudotuberculosis* was grown under anaerobic conditions in M9 supplemented with nitrate and mannitol instead of glucose to support anaerobic respiration. Cultures were then shifted to 37˚C and both secreted and intracellular proteins were analyzed by Western blot, as in Fig 5. Two additional independent replicates are shown. Top panels, supernatant. Bottom panels, cell pellet.
(TIF)

## Acknowledgments

We thank Todd Lowe and Chad Saltikov for help with the vinyl anaerobic chamber, Fitnat Yildiz for advice on experimental design, Hanh Lam for assistance with data analysis, Ana Gallego for advice on RNA extraction optimization, and Virginia Miller for technical advice on construction of the *Y. pestis* Δ*iscR* mutant.

## Author Contributions

**Conceptualization:** Diana Hooker-Romero, Leah Schwiesow, Patricia J. Kiley, Victoria Auerbuch.

**Data curation:** Diana Hooker-Romero.

**Formal analysis:** Diana Hooker-Romero, Erin Mettert.

**Funding acquisition:** Diana Hooker-Romero, Patricia J. Kiley, Victoria Auerbuch.

**Investigation:** Diana Hooker-Romero, Erin Mettert, Leah Schwiesow, David Balderas, Pablo A. Alvarez, Anadin Kicin, Azuah L. Gonzalez.

**Methodology:** Diana Hooker-Romero, Erin Mettert, Leah Schwiesow.

**Project administration:** Victoria Auerbuch.

**Resources:** Gregory V. Plano.

**Supervision:** Patricia J. Kiley, Victoria Auerbuch.

**Validation:** Diana Hooker-Romero.

**Visualization:** Diana Hooker-Romero.

**Writing – original draft:** Diana Hooker-Romero, Victoria Auerbuch.

**Writing – review & editing:** Diana Hooker-Romero, Erin Mettert, Gregory V. Plano, Patricia J. Kiley, Victoria Auerbuch.

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
