## [Decision Letter · Decision Letter 0]

27 Aug 2019

Dear Dr. Auerbuch,

Thank you very much for submitting your manuscript "Iron availability and oxygen tension regulate the Yersinia Ysc type III secretion system through IscR to enable disseminated infection" (PPATHOGENS-D-19-01337) for review by PLOS Pathogens. Your manuscript was fully evaluated at the editorial level and by independent peer reviewers. The reviewers appreciated the attention to an important problem, but raised some substantial concerns about the manuscript as it currently stands. These issues must be addressed before we would be willing to consider a revised version of your study. We cannot, of course, promise publication at that time.

We therefore ask you to modify the manuscript according to the review recommendations before we can consider your manuscript for acceptance. Your revisions should address the specific points made by each reviewer.

(1) A letter containing a detailed list of your responses to the review comments and a description of the changes you have made in the manuscript. Please note while forming your response, if your article is accepted, you may have the opportunity to make the peer review history publicly available. The record will include editor decision letters (with reviews) and your responses to reviewer comments. If eligible, we will contact you to opt in or out.

(2) Two versions of the manuscript: one with either highlights or tracked changes denoting where the text has been changed; the other a clean version (uploaded as the manuscript file).

Additionally, to enhance the reproducibility of your results, PLOS recommends that you deposit your laboratory protocols in protocols.io, where a protocol can be assigned its own identifier (DOI) such that it can be cited independently in the future. For instructions see http://journals.plos.org/plospathogens/s/submission-guidelines#loc-materials-and-methods

We hope to receive your revised manuscript within 60 days. If you anticipate any delay in its return, we ask that you let us know the expected resubmission date by replying to this email. Revised manuscripts received beyond 60 days may require evaluation and peer review similar to that applied to newly submitted manuscripts.

[LINK]

Sincerely,

Joan Mecsas

Associate Editor

PLOS Pathogens

Nina Salama

Section Editor

PLOS Pathogens

Kasturi Haldar

Editor-in-Chief

PLOS Pathogens

orcid.org/0000-0001-5065-158X

Grant McFadden

Editor-in-Chief

PLOS Pathogens

orcid.org/0000-0002-2556-3526

The manuscript was very favorably received by all three expert reviewers although all three reviewers had suggestions to improve the clarity of the figures, to place the work in the broader perspective of Yersinia pathogenic lifestyle and to emphasize the novelty of your study. Please address the experimental concerns of reviewer one, present a clearer figure 6 (what are the bands in the various lanes and what are empty lanes.). In addition, please respond to all of the comments from the reviewers (major and minor issues).

Reviewer's Responses to Questions

**Part I - Summary**

Reviewer #1: The manuscript by Hooker-Romero et al. focuses on the mechanisms that control type-III secretion system (T3SS) expression in Y. pseudotuberculosis, specifically the oxygen and iron-dependent regulation of the system by IscR. The authors show that Y. pseudotuberculosis IscR regulates T3SS expression in response to oxygen and iron availability, and using a mutant that lacks IscR binding to the lcrF promoter, they show this is dependent on lcrF promoter binding. T3SS expression was induced under aerobic conditions, and also anaerobic, iron-depleted conditions. Overall, the novelty of the study could have been better defined, which lessened the enthusiasm for this manuscript. Although oxygen has not been shown to impact T3SS expression in Yersinia, this has been shown in other bacterial species, and Fe-dependent regulation of the T3SS has been shown previously by this research group. Therefore, I believe the novelty of the study is linking both of these environmental signals to IscR-dependent regulation of the T3SS, at the level of lcrF promoter activity. The methodology of this study was sufficiently rigorous, and the evidence justifies the authors’ conclusions.

Reviewer #2: IscR has previously been shown to regulate the expression of the Yersinia Ysc T3SS. Here, Hooker-Romero et al. expand on these initial observations to show that expression of the T3SS can be affected by both oxygen and iron availability, two signals that have not been previously linked to T3SS regulation. Moreover, they show that this regulation occurs through IscR interactions with the promoter of the T3SS master regulator LcrF. Using a combination of gene expression, biochemical, and animal models, they also demonstrate that IscR-binding to the lcrF promoter is required for proper regulation of the T3SS during infection. These data provide a significant improvement in our understanding of how Yersinia senses its environment through IscR to modulate the expression of the T3SS at appropriate times and locations during infection to maximize fitness and virulence potential.

The integration of oxygen and iron into the regulation of the T3SS has not been described before, and these data provide important advances in our understanding of the mechanisms used by Yersinia to coordinate the expression of its major virulence factor to ensure expression within the proper location to maximize virulence. Approaches were rigorous and included appropriate controls. No major issues were observed.

Reviewer #3: Many bacterial pathogens use the availability of iron and iron-sensing proteins to control expression of virulence factors, illustrating the global importance to study these systems. The present study by Hooker-Romeo et al. investigates the regulation of Ysc-type III secretion system (T3SS) of Yersinia pseudotuberculosis by the 2Fe-2S containing regulator IscR via the T3SS regulator LcrF. IscR was already shown to control ysc-T3SS and to be important for virulence in their previous study, but this work adds information about how this regulator responds to iron and oxygen availability. Using gene fusions and westernblot analysis they could demonstrate that T3SS is expressed at both aerobic and anaerob growth conditions through activation by IscR, but under anaerobic condition iron depletion or limitation is required for induction. Moreover, they demonstrate that a non-iscR-binding promoter mutation of lcrF influences virulence, in particular colonization of liver and spleen, very similar to an lcrF mutant.

**Part II – Major Issues: Key Experiments Required for Acceptance**

Reviewer #1: Additional Experiments:

• The experiment with Y. pestis (Figure 8) was performed under different conditions (+/- Ca2+) to induce the T3SS; an additional experiment with Y. pseudotuberculosis performed in the same way is needed to directly compare between Y. pestis and Y. pseudotuberculosis.

• The Apo-IscR mutant was tested under both iron replete and iron limited conditions (line 253) for the anaerobic respiration conditions (Figure 5), but was not tested under iron limited conditions in Figure 6. Was there a reason this wasn’t included? I think this is needed to support the connection between IscR levels and T3SS induction (lines 266-267), since the Apo-IscR mutant is also lysing under anaerobic conditions (lines 260-263).

Major Concerns:

• It was unclear what is known about the requirements for T3SS expression during enteropathogenic Yersinia infection. The abstract suggests that T3SS expression is required for dissemination, but not required for colonization of the intestine; is there a requirement for expression in Peyer’s patches? The results presented here suggest expression is only required for dissemination, was this a novel finding? Or is this consistent with published literature? A discussion of previous studies with strains lacking the virulence plasmid, or lacking T3SS function specifically, would help put this study in context.

• If both Y. pseudotuberculosis and Y. pestis utilize a similar mechanism for T3SS expression, then does this suggest T3SS expression is most important in a niche they both occupy? Such as lymph nodes or deep tissues? More discussion should be included to consider whether T3SS expression is needed within distinct tissues, such as mesenteric lymph nodes. Do these results suggest Y. pestis is still adapted to intestinal tissues, and may not be regulating T3SS expression as efficiently as it could?

• Anatomically, where you would expect the system to be expressed, given the in vitro results presented here? It was unclear which tissues are predicted to have low or higher iron, and which would have low or higher oxygen levels. It would be helpful to mention that culture conditions are meant to mimic specific tissue sites (low iron, aerobic conditions are expected to mimic deep tissues, etc), to help better explain the expected and observed results in the oral infection model.

• The introduction focuses on iron homeostasis, iron availability, and iron-dependent regulation of virulence factors, but doesn’t mention what is known about oxygen-dependent regulation in Yersinia or other bacterial species. A paragraph about oxygen-dependent regulation of virulence factor expression, and specifically the T3SS, is needed to help determine the gaps in this literature, and how this study addresses those gaps. An extended discussion about T3SS expression, and what is known about environmental triggers in general, would also be really helpful.

• The novelty of this study needs to be outlined more clearly. For example, lines 130-133 state that it is known IscR is essential for T3SS expression, is the link with iron availability already known? Or was this unclear prior to this study? What was known about oxygen-dependent regulation of T3SS in this bacterium, or in other bacterial species? Has there been a link between IscR-dependent T3SS expression and oxygen, or is this a novel aspect of this study?

• Densitometry is needed throughout to determine differences in protein levels relative to loading controls. This quantification was mentioned in the text describing Figure 6, but the quantification should be indicated on the blots throughout. This would also help with the interpretation for Figure 3, where it was difficult to determine if mRNA levels were different than protein levels.

• Changes in T3SS expression are described differently in the text than are shown in the figures; the figures show normalized data, while the text describes fold changes. These should match, and I think the fold changes are more helpful in visualizing induction.

• It was unclear why the authors thought anaerobic respiration would alter IscR-dependent phenotypes (Figure 5). The authors found the same result under these conditions, but it was unclear why this could have had an impact.

• A recent Salmonella study showed that IscR regulates T3SS expression, although the conditions that promote expression (high iron, low oxygen) contrasted with the Yersinia conditions that promote T3SS expression (low iron, low oxygen). This recent study was mentioned at the end of the discussion, but should also have been mentioned in the introduction to this study, in addition to any evidence that may suggest the IscR-dependent regulation differs in the two species. Maybe the promoter binding motifs are slightly different? Maybe these systems have adapted differently for intracellular vs. extracellular growth?

Reviewer #2: No major issues

Reviewer #3: The IscR protein is conserved among enteric Gram-negative pathogens and the role of IscR as a sensor of Fe-S cluster demand, which depends on iron and oxygen has also been studied in Salmonella, Pseudomonas aeruginosa and E. coli, making the observation that IscR regulates T3SS in response to iron and oxygen less surprising. However, the overall outcome of IscR-mediated control on T3SS can be very different, indicating that this system co-evolved to adjust expression of these important virulence traits to the bacteria-specific colonized niches. The author should/could more emphasize the novelty of their study.

Overall the authors address a topic that is interesting for our understanding of the role of the Isc regulator family for virulence gene control. The data convincingly show that IscR binding on the lcrF promoter is critical for T3SS and depends on oxygen and iron availability, but the quality of some of the figures could be improved (for detailed comments see below).

Fig. 2B: The authors clearly showed that iscR expression is clearly induced at 37°C under aerobic conditions, but the regulatory mechanism remains unknown.

**Part III – Minor Issues: Editorial and Data Presentation Modifications**

Reviewer #1: Minor comments:

• Author Summary (lines 55-58): this described the varied environments enteropathogenic Yersinia would experience during infection, but what about Y. pestis? Does it also encounter an iron-replete, low oxygen environment like the intestine? This information could be included here, but could also be included at multiple points in the manuscript where comparisons between different Yersinia are made.

• Lines 74-77: emphasize this is prior to infection.

• Line 81: is lipocalin-2 also expressed by the epithelium? Please include a citation here (line 82).

• Line 87-88: consider adding some specific examples of uptake systems.

• Some of the information in the Introduction could be expanded to provide more specific examples: Line 90: which environments, which iron sources? Line 95: which bacterial species? Line 96: which transcription factors? Line 116: which cellular processes? Line 122-123: which Gram-negative organisms, which genes?

• Lines 111-113: is it known which form of IscR binds to the lcrF promoter? Which motif is upstream of lcrF? Is this a gap that will be addressed in this study?

• Line 127: which form of IscR?

• Lines 149-155: It was unclear why the authors wanted to establish the growth phenotype of the lcrFpNull mutant.

• Line 186: typo, ‘p= 0.0 005’.

• Line 234: typo, ‘amount’ should be ‘amounts’

• Line 247-248: are there other expected alterations in the Apo-IscR strain?

• Line 371-374: Does the ∆iscR strain colonize Peyer’s patches? Does this suggest the T3SS is required to colonize all the listed tissues?

• Line 393-396: this section was confusing, since it was mentioned earlier that the T3SS isn’t required for mesenteric lymph node colonization. Does this mean the system is expressed within lymph nodes, but isn’t needed?

• Line 408: typo, ‘pneumonia’ should be ‘pneumoniae’

• Line 585-586: ‘Bases previously shown to be important for IscR binding are in bold’, the bases weren’t in bold.

• Figure 7: significance bar for Peyer’s patches needs to shift slightly to the right

Reviewer #2: Minor issues:

1. Line 206: Check coma in “p<0,0001”

2. Line 586; Figure 1: Does not appear that nts important for binding are not in bold as suggested in figure legend.

3. Line 605; Figure 2: Figure legend shows p value for ****, but there is only *** in the figure. Please make sure values match up.

4. Line 620; Figure 3: Please make sure p values in the figure legend match up with the figure.

5. Line 641; Figure 3: Please make sure p values in the figure legend match up with the figure.

6. Line 674: Please clarify if this is a representative of 5 independent experiments or combined data from 5 experiments.

7. Figure 7: Please denote the limit of detection on each graph. Since your data is represented as CFU per gram, the limit of detection should be different for each tissue (the spleen will impact total volume more than the PP, and so will affect the primary dilution factor in the back calculation differently). Also, for samples below the limit of detection, it is more appropriate to set the CFU for these samples to the limit of detection for each tissue prior to calculating the mean and performing statistics.

Reviewer #3: l. 143: (12) should be (12)

l. 144: The unpublished observations should be more specified to illustrate that IscR is a global regulator

l. 186: P=0.0 005 should be P=0.0005

l. 248-253: This part seems contradictory or did I get this wrong?

l. 249….the Apo-IscR mutant expressed LcrF, but was unable to secrete Yops due to a proton motive force defect….. l. 250 the Apo-IscR mutant had high IcsR levels..this correlated with high levels of LcrF and secreted YopE ….????

Fig. 1A

An unspecific labelled DNA fragment or unlabeled competitive DNA would be helpful to distinguish between specific and unspecific binding. The presented gel shift assays are not fully convincing, as defined protein-complexes also appear with the lcrFPnull fragment.

Fig. 3 and Fig. 4

I suggest to add aerobic on top of Fig. 3 and anaerobic on top of Fig. 4 to make it easier for the reader to see which conditions were used.

Fig. 4A

The data presented demonstrate that even in the absence of the IscR protein, lcrF and yopE expression are upregulated in the absence of iron, although the overall level is reduced. Is the expression of other regulators, e.g. YmoA or RcsB influenced by iron under these conditions?

Fig. 4B

Labelling of YopE and YopD need to be exchanged according to Fig. 3C.

Fig. 5 and 6

The different bands are not well defined due to different sizes and spillover of the slots and could be improved to exclude flow-over. In Fig. 6 it is further unclear what is loaded on lane 2? 5 lanes are visible, but only 4 are labelled. I assume the second lane was empty and the visible bands result from band 1 and 3.

Fig. 8

The unspecific band just above the YopE band should be indicated as unspecific.

PLOS authors have the option to publish the peer review history of their article (what does this mean?). If published, this will include your full peer review and any attached files.

Reviewer #1: No

Reviewer #2: No

Reviewer #3: No

---

## [Editor Report · Decision Letter 1]

10 Nov 2019

Dear Dr. Auerbuch,

We are pleased to inform that your manuscript, "Iron availability and oxygen tension regulate the Yersinia Ysc type III secretion system to enable disseminated infection", has been editorially accepted for publication at PLOS Pathogens. 

Before your manuscript can be formally accepted and sent to production, you will need to complete our formatting changes, which you will receive by email within a week. Please note that your manuscript will not be scheduled for publication until you have made the required changes.

IMPORTANT NOTES

(1) Please note, once your paper is accepted, an uncorrected proof of your manuscript will be published online ahead of the final version, unless you’ve already opted out via the online submission form. If, for any reason, you do not want an earlier version of your manuscript published online or are unsure if you have already indicated as such, please let the journal staff know immediately at plospathogens@plos.org.

(2) Copyediting and Proofreading: The corresponding author will receive a typeset proof for review, to ensure errors have not been introduced during production. Please review the PDF proof of your manuscript carefully, as this is the last chance to correct any errors. Please note that major changes, or those which affect the scientific understanding of the work, will likely cause delays to the publication date of your manuscript. 

(3) Appropriate Figure Files: Please remove all name and figure # text from your figure files. Please also take this time to check that your figures are of high resolution, which will improve the readbility of your figures and help expedite your manuscript's publication. Please note that figures must have been originally created at 300dpi or higher. Do not manually increase the resolution of your files. For instructions on how to properly obtain high quality images, please review our Figure Guidelines, with examples at: http://journals.plos.org/plospathogens/s/figures.

(4) Striking Image: Please upload a striking still image to accompany your article if one is available (you can include a new image or an existing one from within your manuscript). Should your paper be accepted, this image will be considered for our monthly issue image and may also appear on our website to feature your article. Please upload this as a separate file, selecting "striking image" as the file type upon upload. Please also include a separate "Other" file with a caption, including credits and any potential copyright information. Please do not include the caption in the main article file. If your image is from someone other than yourself, please ensure that the artist has read and agreed to the terms and conditions of the Creative Commons Attribution License at http://journals.plos.org/plospathogens/s/content-license. Please note that PLOS cannot publish copyrighted images.

(5) Press Release or Related Media: If your institution or institutions have a press office, please notify them about your upcoming paper at this point, to enable them to help maximize its impact. If they will be preparing press materials for this manuscript, please inform our press team in advance at plospathogens@plos.org as soon as possible. We ask that you contact us within one week to plan ahead of our fast Production schedule. If you need to know your paper's publication date for related media purposes, you must coordinate with our press team, and your manuscript will remain under a strict press embargo until the publication date and time. This means an early version of your manuscript will not be published ahead of your final version. 

(6)  PLOS requires an ORCID iD for all corresponding authors on papers submitted after December 6th, 2016. Please ensure that you have an ORCID iD and that it is validated in Editorial Manager.  To do this, go to ‘Update my Information’ (in the upper left-hand corner of the main menu), and click on the Fetch/Validate link next to the ORCID field.  This will take you to the ORCID site and allow you to create a new iD or authenticate a pre-existing iD in Editorial Manager

(7) Update your Profile Information: Now that your manuscript has been provisionally accepted, please log into Editorial Manager and update your profile, if needed. Go to https://www.editorialmanager.com/ppathogens, log in, and click on the "Update My Information" link at the top of the page. Please update your user information to ensure an efficient production and billing process. 

(8) LaTeX users only: Our staff will ask you to upload a TEX file in addition to the PDF before the paper can be sent to typesetting, so please carefully review our Latex Guidelines http://journals.plos.org/plospathogens/s/latex in the meantime.

(9) If you have associated protocols in protocols.io, please ensure that you make them public before publication to guarantee immediate access to the methodological details.

Best regards,

Joan Mecsas

Associate Editor

PLOS Pathogens

Nina Salama

Section Editor

PLOS Pathogens

Kasturi Haldar

Editor-in-Chief

PLOS Pathogens

orcid.org/0000-0001-5065-158X

Grant McFadden

Editor-in-Chief

PLOS Pathogens

orcid.org/0000-0002-2556-3526
---

## [Editor Report · Acceptance letter]

18 Dec 2019

Dear Dr. Auerbuch,

We are delighted to inform you that your manuscript, "Iron availability and oxygen tension regulate the Yersinia Ysc type III secretion system to enable disseminated infection," has been formally accepted for publication in PLOS Pathogens.

Best regards,

Kasturi Haldar

Editor-in-Chief

PLOS Pathogens

orcid.org/0000-0001-5065-158X

Grant McFadden

Editor-in-Chief

PLOS Pathogens

orcid.org/0000-0002-2556-3526